# Different Tyrosine Kinase Inhibitors Used in Treating EGFR-Mutant Pulmonary Adenocarcinoma with Brain Metastasis and Intracranial Intervention Have No Impact on Clinical Outcomes

**DOI:** 10.3390/cancers15010187

**Published:** 2022-12-28

**Authors:** Chia-Yu Kuo, Ming-Ju Tsai, Jen-Yu Hung, Kuan-Li Wu, Ying-Ming Tsai, Yu-Chen Tsai, Cheng-Hao Chuang, Tai-Huang Lee, Huang-Chi Chen, Chih-Jen Yang, Inn-Wen Chong

**Affiliations:** 1Division of Pulmonary and Critical Care Medicine, Department of Internal Medicine, Kaohsiung Medical University Hospital, Kaohsiung Medical University, Kaohsiung 80708, Taiwan; 2Division of Pulmonary and Critical Care Medicine, Department of Internal Medicine, Kaohsiung Municipal Siaogang Hospital, Kaohsiung 81267, Taiwan; 3School of Medicine, College of Medicine, Kaohsiung Medical University, Kaohsiung 80708, Taiwan; 4Department of Internal Medicine, Kaohsiung Municipal Ta-Tung Hospital, Kaohsiung Medical University, Kaohsiung 80145, Taiwan; 5School of Post-Baccalaureate Medicine, College of Medicine, Kaohsiung Medical University, Kaohsiung 80708, Taiwan; 6Department of Respiratory Therapy, College of Medicine, Kaohsiung Medical University, Kaohsiung 80708, Taiwan

**Keywords:** epidermal growth factor receptor, tyrosine kinase inhibitor, brain metastasis, lung cancer, adenocarcinoma

## Abstract

**Simple Summary:**

Brain metastasis is a factor of a poor prognosis in patients with non-small-cell lung cancer (NSCLC) harboring epidermal growth factor receptor (EGFR) mutations. The standard treatment is systemic therapy combined with intracranial intervention, such as craniotomy or radiotherapy. However, intracranial intervention may result in neurological or cognitive deficiency. We conducted a retrospective study to determine the optimal treatment strategy for EGFR-mutant NSCLC patients with brain metastasis receiving or not receiving intracranial intervention. Intracranial intervention had no statistically significant impact on response rate (RR), progression-free survival (PFS), or overall survival (OS) of patients with EGFR mutations and brain metastasis who received EGFR tyrosine kinase inhibitors (TKIs) as a first-line therapy. Treatment with different EGFR TKIs did not result in significant differences in RR or OS, but PFS differed significantly between the therapies. Afatinib and osimertinib both showed significantly longer PFS than gefitinib in a Cox regression model. The OS of patients with higher graded prognostic assessment (GPA) scores was significantly longer. Physicians should be vigilant about patients who have lower GPA scores at initial diagnosis of NSCLC with brain metastasis.

**Abstract:**

Brain metastasis in patients with non-small-cell lung cancer (NSCLC) harboring epidermal growth factor receptor (EGFR) mutations is a factor of poor prognosis. We conducted a retrospective study to determine the optimal treatment strategy for EGFR-mutant NSCLC patients with brain metastasis receiving or not receiving intracranial intervention. A total of 186 patients treated with an EGFR TKI were enrolled in the study, and 79 (42%) received intracranial intervention. Patients who received intracranial intervention and those who did not had a similar treatment response rate (RR), progression-free survival (PFS) (median PFS: 11.0 vs. 10.0 months, *p* = 0.4842), and overall survival (OS) (median OS: 23.0 vs. 23.2 months, *p* = 0.2484). Patients treated with gefitinib, erlotinib, afatinib, or osimertinib had a similar RR (63%, 76%, 81%, or 100%, respectively, *p* = 0.1390), but they had significantly different PFS (median PFS: 7.5, 10.0, 14.8 months, or not reached, respectively, *p* = 0.0081). In addition, OS tended to be different between different EGFR TKI treatments (median OS of 19.2, 23.7, or 33.0 months for gefitinib, erlotinib, or afatinib treatments, respectively, *p* = 0.0834). Afatinib and osimertinib both demonstrated significantly longer PFS than gefitinib in a Cox regression model. Graded prognostic assessment (GPA) versions 2017 and 2022 stratified patients with different OS; patients with higher GPA index scores had significantly longer OS (*p* = 0.0368 and 0.0407 for version 2017 and 2022, respectively).

## 1. Introduction

Non-small-cell lung cancer (NSCLC) is the most common cause of cancer-related mortality worldwide, and the most prevalent cell type of this cancer is adenocarcinoma [1]. Brain metastasis is common in advanced NSCLC. Approximately 10–20% of patients have brain metastasis at initial diagnosis of NSCLC, and about 20–40% develop brain metastasis during treatment [2]. The standard treatment for NSCLC patients with brain metastasis is chemotherapy combined with intracranial intervention, for example, craniotomy or radiotherapy, beforehand [3]. However, not all NSCLC patients with brain metastasis are suitable for surgery due to factors such as poor performance status, tumor occurrence in an inoperable site, and multiple brain metastases [4]. Although the rate of incidence of surgical complications is not high as previously thought, careful, shared decision-making about surgery is still necessary. Some NSCLC patients with brain metastasis are treated with radiotherapy, but previous reports show poor prognoses and a median survival of about 6 months [5]. In addition, NSCLC patients with brain metastasis may show poor responsiveness to chemotherapy, including poor response rates (27–69%) and a short duration of overall survival (OS) (7.4–10 months) [6,7,8]. 

In the era of precision medicine, the development of epidermal growth factor receptor (EGFR) tyrosine kinase inhibitors (TKIs) has changed clinical practice. The rate of incidence of the EGFR mutation in NSCLC patients is approximately 40–60% among Asian and 10% among Western patients [9]. In recent years, randomized trials have further shown that patients with NSCLC harboring susceptible EGFR mutations, such as the exon 21 L858R point mutation and exon 19 deletion, have longer progression-free survival (PFS) when they are treated with EGFR TKIs rather than standard chemotherapy [10,11,12,13,14,15,16]. Previous studies have also reported encouraging results for EGFR TKI treatment of patients with EGFR mutations and brain metastasis [10,13,14,15,17,18,19].

EGFR TKIs have different in vitro sensitivities, plasma drug concentrations, clinical responses, and rates of penetration into the cerebrospinal fluid from the plasma [20,21,22,23,24]. According to previous studies which compared the treatment outcomes of different EGFR TKIs for EGFR-mutant NSCLC patients with brain metastasis, different first-line EGFR TKIs achieved similar treatment outcome [25]. However, the clinical efficacy of different EGFR-TKI-only treatments for patients with brain metastasis is still unknown. In addition, EGFR TKIs combined with brain surgery or radiotherapy to treat metastatic brain tumors is still a popular strategy for patients with EGFR mutations and brain metastasis. However, memory impairment is inevitable in NSCLC patients with brain metastasis after whole-brain radiotherapy [26]. The role of surgery or radiotherapy in treating patients with susceptible EGFR mutations and brain metastasis is still unclear. We conducted this retrospective study with real-world data from three hospitals to determine the optimal treatment strategy for EGFR-mutant NSCLC patients with brain metastasis. 

## 2. Materials and Methods

### 2.1. Patient Identification

We enrolled patients with lung adenocarcinoma with brain metastasis diagnosed and treated at three hospitals affiliated with Kaohsiung Medical University—Kaohsiung Medical University Hospital, Kaohsiung Municipal Ta-Tung Hospital, and Kaohsiung Municipal Siaogang Hospital (Figure 1). The diagnosis of lung adenocarcinoma was confirmed pathologically according to the World Health Organization pathology classification, and cancer stage was confirmed by a lung cancer team according to the 8th version of the American Joint Committee on Cancer staging system. Formalin-fixed paraffin-embedded tissue blocks were made from tumor specimens obtained from the enrolled patients. Genomic DNA was extracted from tissue blocks for genotyping exons 18 to 21 of the EGFR gene with the cobas^®^ EGFR Mutation Test v2 (Roche Molecular Systems, Inc., Basel, Switzerland). The ready-to-use test kit allows the detection of 42 somatic mutations in the cancer-related EGFR gene with a polymerase chain reaction using a cobas^®^ z480 instrument and a cobas^®^ 4800 analyzer. The examination techniques were the same as those in our previous studies [27,28,29,30]. The Institutional Review Board (IRB) of Kaohsiung Medical University Hospital (KMUH) approved this study (KMUHIRB-E(I)-20210257) and waived the need for written informed consent from all patients.

In the present study, we enrolled individuals with only the exon 19 deletion or exon 21 L858R point mutation in lung adenocarcinoma. All patients were naïve to systemic treatment and received an EGFR TKI—gefitinib, erlotinib, afatinib, or osimertinib—as their first-line systemic treatment. The patients’ medical records were retrospectively reviewed to determine their baseline clinical characteristics: age at diagnosis, sex, Eastern Cooperative Oncology Group (ECOG) performance status, Karnofsky performance status, EGFR mutation, programmed death ligand 1 (PD-L1), number of metastatic brain nodules, site of extracranial metastasis (ECM), and anti-VEGF treatment. Patients receiving intracranial intervention, i.e., surgery or radiotherapy, for a metastatic brain tumor were identified. Graded prognostic assessment (GPA) index scores (Table A1) were calculated according to three previously published versions—2010 [31], 2017 [32], and 2022 [33].

The response to the initial treatment was classified based on serial imaging studies using the revised Response Evaluation Criteria in Solid Tumors v. 1.1 (RECIST). PFS, intracranial PFS (iPFS), and OS were defined as the time from the start of the initial EGFR TKI treatment to the date of disease progression detected with an imaging examination, the date of disease progression detected on a brain image, and the date of death, respectively. 

### 2.2. Statistical Analysis

In this retrospective study, we adopted the statistical approaches used in our previous studies [27,28,29,30] and in other similar studies [25]. Categorical variables and continuous variables were compared using a chi-squared test and analysis of variance, respectively. Survival times were estimated using the Kaplan–Meier method, and differences between groups were compared with the log-rank test. A Cox regression analysis was used to determine the predictive factors of PFS, iPFS, and OS. Both univariate and multivariable analyses were performed. To avoid overadjustment, we used a backward variable selection method, retaining only variables with a *p*-value of < 0.15, so as to develop reduced multivariate models. Hazard ratios (HR) with 95% confidence intervals (CIs) are presented for the predictors. All statistical analyses were performed with SAS software (version 9.4 for Windows, SAS Institute Inc., Cary, NC, USA). A two-tailed *p*-value of < 0.05 was set as the critical value for statistical significance.

## 3. Results

### 3.1. Study Characteristics 

We identified 193 patients with lung adenocarcinoma and brain metastasis treated with an EGFR TKI as their first-line systemic treatment (Figure 1). After excluding those with the exon 18 or 20 mutation or a known resistant mutation, the remaining 186 patients with the exon 19 deletion or exon 21 L858R point mutation were enrolled in the study (Table 1 and Table A1). The mean (±standard deviation) age of the enrolled patients was 65.5 (±11.2) years, and 63 (34%) patients were male. Most patients’ performance status upon diagnosis was good. (A total of 85% had an ECOG performance status of ≤1, and 84% had a Karnofsky performance status of ≥80.) The exon 19 deletion and exon 21 L858R were detected in the tumors of 85 (46%) and 102 (55%) of the patients, respectively (one patient’s tumor had both mutations). A total of 63 (34%) patients had a single brain metastasis, while 68 (37%) had more than four metastatic brain nodules. 

### 3.2. Intracranial Intervention

Of the 186 patients enrolled in this study, 79 (42%) received intracranial intervention for a metastatic brain tumor. (A total of 50 patients underwent surgery, and 61 underwent radiotherapy) (Table A2). Patients receiving intracranial intervention tended to have fewer brain metastases (*p* = 0.0600) and significantly fewer ECM (*p* = 0.0174) than those who did not receive intracranial intervention, while the two groups’ performance statuses were similar. The patients receiving intracranial intervention and those who did not receive the intervention showed a similar response to the initial treatment (Table A3), and the PFS (median PFS: 11.0 vs. 10.0 months; *p* = 0.4842) and OS (median OS: 23.0 vs. 23.2 months; *p* = 0.2484) of the two groups were also similar (Figure 2A,B and Figure A1A,B).

### 3.3. Different EGFR TKIs

EGFR TKIs taken as the first-line systemic treatment were gefitinib (27 patients), erlotinib (104 patients), afatinib (48 patients), and osimertinib (7 patients) (Table 1). Patients receiving different EGFR TKIs were similar in age, performance status, and number of ECM. Patients with a higher number of metastatic brain nodules were less likely to be treated with gefitinib. The initial treatment response rate was similar in patients treated with gefitinib, erlotinib, afatinib, and osimertinib (63%, 76%, 81%, and 100%, respectively; *p* = 0.1390) (Table 2). The PFS of patients taking gefitinib, erlotinib, afatinib, or osimertinib was significantly different (median PFS: 7.5 months, 10.0 months, 14.8 months, or not reached, respectively; *p* = 0.0081) (Figure 2C). A similar result was noted for patients without intracranial intervention (Figure A2A), but not for those with intracranial intervention (Figure A2C). The OS of patients taking gefitinib, erlotinib, afatinib, or osimertinib tended to be different (median OS: 19.2 months, 23.7 months, and 33.0 months, or not reached, respectively; *p* = 0.0834) (Figure 2D), while these results were not repeated after the patients were stratified by whether or not they received intracranial intervention (Figure A2B,D).

### 3.4. Prognostic Factors for PFS and OS

Univariate Cox regression analyses showed that treatment with afatinib or osimertinib was significantly associated with better PFS (HR [95% CI] = 0.54 [0.33–0.88], *p* = 0.0133 for afatinib; HR [95% CI] = 0.10 [0.01–0.75], *p* = 0.0248 for osimertinib) (Table 3), whereas male sex (HR [95% CI] = 1.54 [1.09–2.17], *p* = 0.0133) and bone metastasis (HR [95% CI] = 1.57 [1.12–2.21], *p* = 0.0091) were significantly associated with poorer PFS. Multivariable analyses with backward variable selection confirmed that afatinib and osimertinib were independent prognostic factors of better PFS, while male sex and bone metastasis were independent prognostic factors of poorer PFS (model 2R in Table 3). Univariate Cox regression analyses or multivariable analyses did not find any factors significantly associated with iPFS (Table A4).

Univariate Cox regression analyses revealed that the male sex was associated with significantly poorer OS (HR [95% CI] = 1.52 [1.03–2.26], *p* = 0.0368) (Table 4). Multivariate analyses with backward variable selection showed that independent prognostic factors of poorer OS included male sex (HR [95% CI] = 1.63 [1.09–2.44], *p* = 0.0169) and extracranial metastasis (HR [95% CI] = 1.99 [1.14–3.49], *p* = 0.0160) (model 3R in Table 4).

We then examined whether different versions of the GPA (Table A1) could properly stratify patients’ risk. Patients stratified with the 2010, 2017, or 2022 version of the GPA showed a similar PFS (Figure A3A,C,E). Patients stratified with GPA version 2010 had similar OS (Figure A3B). GPA versions 2017 and 2022 appropriately stratified patients with different OS, while patients with higher GPA scores had significantly longer OS (*p* = 0.0368 and *p* = 0.0407 for GPA versions 2017 and 2022, respectively) (Figure A3D,F).

## 4. Discussion

This study made several important findings. First, the patients who received intracranial intervention and those who did not showed a similar response to the initial treatment; these groups were also similar in PFS and OS. Second, patients responded in a similar way to gefitinib, erlotinib, afatinib, and osimertinib as the initial treatment. Third, afatinib and osimertinib were independent prognostic factors of a better PFS, while male sex and bone metastasis were independent prognostic factors of a poorer PFS. The independent prognostic factors of a poorer OS were male sex and extracranial metastasis. Finally, GPA versions 2017 and 2022 appropriately stratified patients with different OS, while patients with higher GPA scores had significantly longer OS.

Brain metastasis is associated with poor prognosis of NSCLC patients [34]. Systemic therapy combined with surgery or radiotherapy is the standard treatment. A previous retrospective study of 296 patients evaluated the benefits of craniotomy for patients with NSCLC and brain metastases. Most of the patients had ≤3 brain lesions, and about 90% of the patients received whole-brain radiotherapy for their brain metastasis. However, the overall survival of the relatively few patients (less than 10%) who underwent craniotomy was significantly longer. Regardless of EGFR mutation status, craniotomy remains a strong prognostic factor of better survival. Some patients may develop complications after craniotomy, the most serious being intracranial hemorrhage, which could cause mortality [35]. In our study, 27% of the patients underwent craniotomy for brain metastasis, but their PFS and OS were similar to those of patients who did not have these interventions.

Radiotherapy for brain lesions is the standard of care for brain metastasis in patients who are not suitable for craniotomy. In a previous retrospective study, which compared the efficacy of first-line EGFR TKIs in combination with radiotherapy vs. EGFR TKI only for patients with EGFR-mutant lung adenocarcinoma with brain metastasis, treatment of symptomatic brain metastasis with the combined therapy achieved a higher response rate and significant improvement in iPFS than did EGFR TKI alone [36]. However, in patients with asymptomatic brain metastasis, the combined EGFR TKI and radiotherapy treatment and EGFR-TKI-only treatment achieved similar iPFS, but the former may result in memory or cognition impairment months later [36]. Another study that evaluated patients with metastatic brain cancer receiving whole-brain radiotherapy showed significant memory impairment after brain radiotherapy. These results suggest that memory impairment may be an early marker of cognitive impairment in patients with brain metastasis who undergo brain radiotherapy [26]. In the present study, 33% of the patients received radiotherapy, which, in almost all cases, was whole-brain radiotherapy. We found that the rate of response to systemic treatment, PFS, and OS of patients receiving radiotherapy were similar to those of patients who did not receive radiotherapy. The intracranial intervention had no statistically significant impact on the survival of the patients with EGFR mutations who received only EGFR TKIs as the first-line therapy.

Although NSCLC progression depends on driver mutations, it is also affected by the extracellular matrix interaction which is mediated by integrins [37,38]. Integrins are cell adhesions and play a key role in the regulation of the process of tumor angiogenesis, which consists of basement membrane degradation and endothelial cell migration, proliferation, and stabilization [39,40]. EGFR regulates integrin activation and cell adhesion, providing control over cellular responses to the environment [41]. Deciphering the molecular mechanisms underlying EGFR and extracellular matrix interactions might provide a better understanding of disease pathobiology and aid in developing therapeutic strategies. According to previous studies, EGFR TKIs are still the standard first-line treatment for EGFR-mutant NSCLC patients. Gefitinib, a first-generation EGFR TKI, has been reported to have low efficacy in treating brain metastasis and poor rates of penetration of the cerebrospinal fluid from the plasma [42]. In contrast, erlotinib, another first-generation EGFR TKI, is effective in treating brain metastasis, is able to cross the blood–brain barrier, and shows a relatively high concentration in the cerebrospinal fluid [43]. Therefore, physicians frequently prescribe erlotinib instead of gefitinib for patients with brain metastasis. Furthermore, the LUX-Lung 3 and LUX-Lung 6 trials showed that afatinib is effective against metastatic brain tumors and yields significantly improved treatment outcomes [13,14]. The FLAURA trial demonstrated that, compared to first-generation TKIs, osimertinib causes a statistically significant improvement in the treatment outcomes of patients with brain metastasis [15,16]. In the subgroup analysis of the central nervous system response in the FLAURA study, osimertinib had significantly better central nervous system efficacy, and patients had a lower intracranial progression rate compared to those who received first-generation EGFR TKIS [44]. The results of the BLOOM study also showed an excellent treatment effect of osimertinib in patients with leptomeningeal carcinomatosis [24]. Chen et al. compared first- and second-generation EGFR TKIs for the treatment of brain metastasis and found no significant differences in PFS, OS, or iPFS in real-world practice between patients who received different generations of EGFR TKIs [25]. In our study, we compared the efficacy between first-, second- and third-generation EGFR TKIs for the treatment of mutant-EGFR lung cancer with brain metastasis. All four EGFR TKIs had similar response rates, but significantly different PFS. Their clinical efficacy was similar in patients without intracranial intervention for brain metastasis. The OS of patients taking different EGFR TKIs was also different. Patients treated with osimertinib had the longest OS compared to those treated with gefitinib, erlotinib, or afatinib. In addition, the OS of patients treated with afatinib was longer than that of patients receiving first-generation EGFR TKIs.

Older age, poor performance status, and extracranial metastasis are regarded as independent factors of poor prognosis of patients with brain metastasis [45]. Chen et al. found that NSCLC patients with uncommon mutations, multiple brain metastases, and concomitant liver metastases tended to have a shorter OS [25]. In our study, multivariable analyses with backward variable selection confirmed that afatinib and osimertinib are independent prognostic factors of better PFS, while male sex and bone metastasis are independent prognostic factors of poorer PFS. Independent prognostic factors of poorer OS included male sex and extracranial metastasis.

GPA has previously been used in decision-making to treat NSCLC patients with brain metastasis [31]. The DS-GPA has been upgraded to the lung-molGPA, which includes the EGFR and ALK mutation status and PD-L1 expression [33]. The lung-molGPA has six factors, with total scores ranging from 0 to 4. Cheng et al. examined the effects of lung-molGPA and different treatment strategies on survival of EGFR-mutant NSCLC patients with brain metastasis; a lung-molGPA of ≥3 was associated with improved OS [46]. In our study, GPA versions 2017 and 2022 appropriately stratified patients with different OS, while patients with higher GPA scores (≥2.5 by version 2017 and ≥3 by version 2022) had significantly longer OS.

Our study has several limitations. First, the number of patients receiving osimertinib was relatively small compared to the number receiving other EGFR TKIs under Taiwan’s National Health Insurance system rules. Taiwan’s National Health Insurance (NHI) is a government-run, single-payer program introduced in 1995 that now covers more than 99% of all Taiwanese citizens. Only gefitinib, erlotinib, and afatinib were allowed by Taiwan’s NHI before 2020. Osimertinib was relatively expensive, and only some patients could afford osimertinib. Osimertinib has been allowed for lung adenocarcinoma with brain metastasis in the deletion 19 subgroup by Taiwan’s NHI since April 2022. This is the reason why only small numbers of patients with lung cancer with brain metastasis were treated with osimertinib in our study. Second, the choice of EGFR TKI for the patient may depend on clinicians’ decisions, which may be influenced by previous studies. In our study, the erlotinib group had significantly more patients than the other groups. Third, the type of radiotherapy for brain tumors in our study was almost always whole-brain radiotherapy. Only a few patients received stereotactic radiosurgery for brain metastasis. Furthermore, iPFS was difficult to evaluate in our study because of its retrospective nature and the check-up interval of the brain image not being clearly defined, which may influence the accuracy of the evaluation of iPFS. 

## 5. Conclusions

In conclusion, our study demonstrated that intracranial intervention had no statistically significant impact on the survival of patients with EGFR mutations and brain metastasis receiving EGFR TKIs as the first-line therapy. In daily practice, many lung cancers with brain metastasis receive intracranial intervention, such as craniotomy and radiotherapy. Our study suggests that intracranial intervention is not necessary for these patients. Patients treated with osimertinib or afatinib had better clinical outcomes compared to those treated with first-generation EGFR TKIs. Afatinib and osimertinib were independent prognostic factors of better PFS. Male sex and bone metastasis were independent prognostic factors of poorer PFS and OS. Patients with higher GPA scores had significantly longer OS. Therefore, physicians should be vigilant about patients with lower GPA scores at the initial diagnosis of NSCLC with brain metastasis.

## Figures and Tables

**Figure 1 cancers-15-00187-f001:**
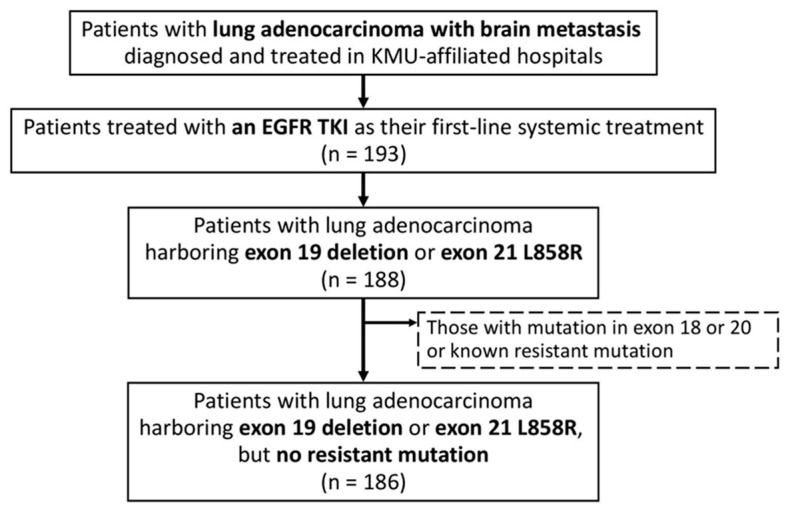
Flowchart for identifying the study population. Abbreviations: EGFR, epidermal growth factor receptor; KMU, Kaohsiung Medical University.

**Figure 2 cancers-15-00187-f002:**
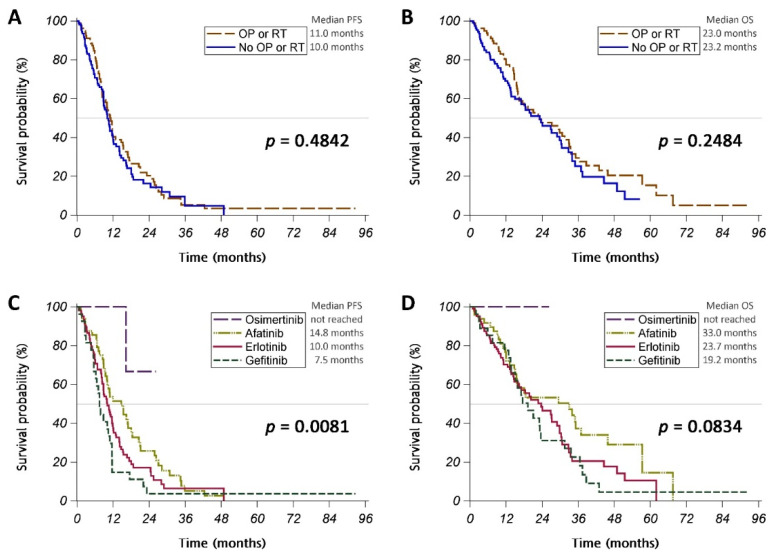
(**A**,**C**) Progression-free survival (PFS) and (**B**,**D**) overall survival (OS) of patients classified according to (**A**,**B**) those undergoing or not undergoing surgery (OP) or radiotherapy (RT) for brain tumor or (**C**,**D**) those receiving different epidermal growth factor receptor tyrosine kinase inhibitors.

**Table 1 cancers-15-00187-t001:** Baseline characteristics of the study cohort.

Variables	All Patients	Gefitinib	Erlotinib	Afatinib	Osimertinib	*p*-Value
n	186	27	104	48	7	
Sex						
Female	123 (66%)	16 (59%)	71 (68%)	30 (63%)	6 (86%)	0.5197
Male	63 (34%)	11 (41%)	33 (32%)	18 (38%)	1 (14%)	
Age (years)	65.5 ± 11.2	64.9 ± 13.2	66.5 ± 10.7	63.3 ± 10.6	68.9 ± 13.6	0.3470
<65	92 (49%)	16 (59%)	47 (45%)	26 (54%)	3 (43%)	0.5044
≥65	94 (51%)	11 (41%)	57 (55%)	22 (46%)	4 (57%)	
ECOG performance status						
0–1	159 (85%)	23 (85%)	90 (87%)	42 (88%)	4 (57%)	0.1884
≥2	27 (15%)	4 (15%)	14 (13%)	6 (13%)	3 (43%)	
Karnofsky performance status						
100	12 (6%)	3 (11%)	7 (7%)	1 (2%)	1 (14%)	0.2980
90	21 (11%)	4 (15%)	9 (9%)	8 (17%)	0 (0%)	
80	125 (67%)	16 (59%)	74 (71%)	32 (67%)	3 (43%)	
70	27 (15%)	4 (15%)	14 (13%)	6 (13%)	3 (43%)	
≤60	1 (1%)	0 (0%)	0 (0%)	1 (2%)	0 (0%)	
EGFR mutation						
Exon 19 deletion	85 (46%)	10 (37%)	46 (44%)	25 (52%)	4 (57%)	0.5591
Exon 21 L858R	102 (55%)	17 (63%)	59 (57%)	23 (48%)	3 (43%)	0.5311
PD-L1 ^†^						
<1%	29 (29%)	1 (33%)	16 (24%)	12 (43%)	0 (0%)	0.1701
≥1%	72 (71%)	2 (67%)	50 (76%)	16 (57%)	4 (100%)	
Number of brain metastases						
1	63 (34%)	15 (56%)	31 (30%)	15 (31%)	2 (29%)	0.0378
2–4	55 (30%)	10 (37%)	30 (29%)	14 (29%)	1 (14%)	
≥5	68 (37%)	2 (7%)	43 (41%)	19 (40%)	4 (57%)	
Number of ECM sites						
0	25 (13%)	4 (15%)	15 (14%)	6 (13%)	0 (0%)	0.6122
1–2	91 (49%)	14 (52%)	45 (43%)	28 (58%)	4 (57%)	
≥3	70 (38%)	9 (33%)	44 (42%)	14 (29%)	3 (43%)	
Site of ECMs						
Lung metastasis	99 (53%)	12 (44%)	58 (56%)	25 (52%)	4 (57%)	0.7590
Pleural metastasis/effusion	81 (44%)	12 (44%)	49 (47%)	17 (35%)	3 (43%)	0.6064
Bone metastasis	116 (62%)	18 (67%)	63 (61%)	30 (63%)	5 (71%)	0.8965
Liver metastasis	26 (14%)	1 (4%)	16 (15%)	7 (15%)	2 (29%)	0.2844
Pericardial metastasis/effusion	51 (27%)	6 (22%)	31 (30%)	11 (23%)	3 (43%)	0.5741
Adrenal metastasis	23 (12%)	2 (7%)	15 (14%)	5 (10%)	1 (14%)	0.7504
Other metastasis	12 (6%)	4 (15%)	6 (6%)	1 (2%)	1 (14%)	0.1424
GPA (version 2010) score						
≥2.5	26 (14%)	7 (26%)	11 (11%)	8 (17%)	0 (0%)	0.2945
1.5–2.0	66 (35%)	11 (41%)	37 (36%)	15 (31%)	3 (43%)	
≤1.0	94 (51%)	9 (33%)	56 (54%)	25 (52%)	4 (57%)	
GPA (version 2017) score						
≥2.5	36 (19%)	8 (30%)	18 (17%)	10 (21%)	0 (0%)	0.0552
1.5–2.0	120 (65%)	19 (70%)	65 (63%)	32 (67%)	4 (57%)	
≤1.0	30 (16%)	0 (0%)	21 (20%)	6 (13%)	3 (43%)	
GPA (version 2022) score						
≥3.0	17 (9%)	3 (11%)	10 (10%)	4 (8%)	0 (0%)	0.1905
1.5–2.5	149 (80%)	24 (89%)	79 (76%)	41 (85%)	5 (71%)	
≤1.0	20 (11%)	0 (0%)	15 (14%)	3 (6%)	2 (29%)	
Anti-VEGF treatment						
No	175 (94%)	27 (100%)	95 (91%)	47 (98%)	6 (86%)	0.1545
Yes	11 (6%)	0 (0%)	9 (9%)	1 (2%)	1 (14%)	
Brain OP (operation for brain tumor)						
No	136 (73%)	16 (59%)	80 (77%)	33 (69%)	7 (100%)	0.0919
Yes	50 (27%)	11 (41%)	24 (23%)	15 (31%)	0 (0%)	
Brain RT (radiotherapy for brain tumor)						
No	125 (67%)	14 (52%)	80 (77%)	24 (50%)	7 (100%)	0.0006
Yes	61 (33%)	13 (48%)	24 (23%)	24 (50%)	0 (0%)	

Abbreviations: ECOG, Eastern Cooperative Oncology Group; EGFR, epidermal growth factor receptor; ECM, extracranial metastasis; GPA, graded prognostic assessment; OP, surgery for brain tumor; PD-L1, programmed death ligand 1; RT, radiotherapy for brain tumor; TKI, tyrosine kinase inhibitor; VEGF, vascular endothelial growth factor. Data are presented as n (%) or mean ± standard deviation. *p*-values are presented for the results of the chi-squared test or the analysis of variance. ^†^ PD-L1 was examined in 101 patients.

**Table 2 cancers-15-00187-t002:** Initial treatment response and site of disease progression.

Variables	All Patients	Gefitinib	Erlotinib	Afatinib	Osimertinib	*p*-Value
Initial treatment response						
Progressive disease (PD)	18 (10%)	4 (15%)	9 (9%)	5 (10%)	0 (0%)	0.6378
Stable disease (SD)	26 (14%)	6 (22%)	16 (15%)	4 (8%)	0 (0%)	
Partial response (PR)	139 (75%)	17 (63%)	77 (74%)	38 (79%)	7 (100%)	
Complete response (CR)	3 (2%)	0 (0%)	2 (2%)	1 (2%)	0 (0%)	
Disease control rate	168 (90%)	23 (85%)	95 (91%)	43 (90%)	7 (100%)	0.6325
Response rate	142 (76%)	17 (63%)	79 (76%)	39 (81%)	7 (100%)	0.1390
Initial intracranial treatment response ^†^						
Intracranial progressive disease (iPD)	9 (6%)	1 (5%)	3 (3%)	5 (11%)	0 (0%)	0.6033
Intracranial stable disease (iSD)	28 (18%)	4 (20%)	19 (22%)	4 (9%)	1 (14%)	
Intracranial partial response (iPR)	76 (48%)	9 (45%)	43 (49%)	20 (45%)	4 (57%)	
Intracranial complete response (iCR)	46 (29%)	6 (30%)	23 (26%)	15 (34%)	2 (29%)	
Intracranial disease control rate	150 (94%)	19 (95%)	85 (97%)	39 (89%)	7 (100%)	0.2667
Intracranial response rate	122 (77%)	15 (75%)	66 (75%)	35 (80%)	6 (86%)	0.8749
Site of disease progression						
Intracranial progression	48 (26%)	5 (19%)	24 (23%)	19 (40%)	0 (0%)	0.0394
Extracranial progression	21 (11%)	5 (19%)	10 (10%)	6 (13%)	0 (0%)	0.4469
Lung	5 (3%)	1 (4%)	3 (3%)	1 (2%)	0 (0%)	0.9439
Pleural nodule/effusion	7 (4%)	3 (11%)	2 (2%)	2 (4%)	0 (0%)	0.1516
Bone	9 (5%)	0 (0%)	6 (6%)	3 (6%)	0 (0%)	0.5455
Liver	2 (1%)	0 (0%)	1 (1%)	1 (2%)	0 (0%)	0.8397
Adrenal gland	1 (1%)	1 (4%)	0 (0%)	0 (0%)	0 (0%)	0.1155
Other	1 (1%)	1 (4%)	0 (0%)	0 (0%)	0 (0%)	0.1155

^†^ Initial intracranial treatment response was assessed in 159 patients. Data are presented as n (%). *p*-values are presented for the results of the chi-squared test.

**Table 3 cancers-15-00187-t003:** Factors associated with progression-free survival.

Variables	Univariate	Multivariable Model 1	Multivariable Model 1R	Multivariable Model 2	Multivariable Model 2R
OP (vs. no OP)	0.90 [0.63–1.28]				
RT (vs. no RT)	0.93 [0.66–1.30]				
OP or RT (vs. no OP or RT)	0.89 [0.65–1.23]	0.81 [0.56–1.15]		0.79 [0.55–1.16]	
EGFR TKI: (vs. gefitinib)					
Erlotinib	0.75 [0.48–1.16]	0.75 [0.44–1.27]	0.79 [0.50–1.23]	0.73 [0.42–1.27]	0.76 [0.49–1.19]
Afatinib	0.54 [0.33–0.88] *	0.52 [0.31–0.89] *	0.53 [0.32–0.87] *	0.51 [0.30–0.88] *	0.51 [0.31–0.84] **
Osimertinib	0.10 [0.01–0.75] *	0.11 [0.01–0.83] *	0.11 [0.01–0.81] *	0.10 [0.01–0.77] *	0.11 [0.02–0.83] *
Anti-VEGF treatment (vs. no anti-VEGF)	0.81 [0.36–1.83]	0.86 [0.37–2.00]		0.94 [0.40–2.22]	
Male (vs. female)	1.54 [1.09–2.17] *	1.52 [1.07–2.17] *	1.50 [1.06–2.11] *	1.64 [1.13–2.39] **	1.57 [1.11–2.23] *
Age (≥65 vs. <65)	0.83 [0.60–1.15]	0.82 [0.57–1.18]		0.87 [0.60–1.27]	
ECOG PS (≥2 vs. ≤1)	0.74 [0.46–1.20]	0.81 [0.48–1.35]		0.86 [0.51–1.45]	
Exon 19 deletion vs. L858R	0.99 [0.72–1.36]	0.87 [0.61–1.23]		0.94 [0.67–1.34]	
PD–L1 ≥ 1% vs. PD-L1 < 1% or not tested	1.00 [0.72–1.40]	1.04 [0.71–1.51]		1.14 [0.77–1.68]	
Number of brain metastases (≥5 vs. <5)	0.87 [0.62–1.22]	1.00 [0.70–1.43]		1.02 [0.70–1.48]	
Extracranial metastasis (vs. no)	1.26 [0.80–1.99]	1.47 [0.91–2.38]	1.40 [0.89–2.22]		
Lung metastasis (vs. no)	1.12 [0.81–1.54]			0.97 [0.67–1.39]	
Pleural metastasis/effusion (vs. no)	1.29 [0.93–1.79]			1.48 [0.98–2.23]	1.38 [0.99–1.92]
Bone metastasis (vs. no)	1.57 [1.12–2.21] **			1.66 [1.14–2.42] **	1.59 [1.13–2.23] **
Liver metastasis (vs. no)	0.99 [0.61–1.61]			0.94 [0.55–1.60]	
Pericardial metastasis/effusion (vs. no)	1.16 [0.80–1.67]			0.93 [0.58–1.49]	
Adrenal metastasis (vs. no)	0.99 [0.59–1.64]			0.87 [0.49–1.55]	
Other site metastasis (vs. no)	0.97 [0.48–1.99]			1.35 [0.61–2.98]	

Abbreviations: ECOG, Eastern Cooperative Oncology Group; EGFR, epidermal growth factor receptor; OP, surgery for brain tumor; PD-L1, programmed death ligand 1; PS, performance status; RT, radiotherapy for brain tumor; TKI, tyrosine kinase inhibitor; VEGF, vascular endothelial growth factor. A patient with cancer harboring both the exon 19 deletion and L858R was included in the exon 19 deletion group. Data are presented as hazard ratio (HR) with 95% confidence interval. After building the maximal models of multivariable Cox regression (models 1 and 2), the corresponding reduced multivariable Cox regression models (models 1R and 2R, respectively) were built using backward variable selection, retaining only variables with *p*-values < 0.15. * *p*-value < 0.05; ** *p*-value < 0.01.

**Table 4 cancers-15-00187-t004:** Factors associated with overall survival.

Variables	Univariate	Multivariable Model 3	Multivariable Model 3R	Multivariable Model 4	Multivariable Model 4R
OP (vs. no OP)	0.80 [0.53–1.21]				
RT (vs. no RT)	0.81 [0.55–1.19]				
OP or RT (vs. no OP or RT)	0.80 [0.55–1.17]	0.89 [0.60–1.33]		0.87 [0.57–1.33]	
EGFR TKI: (vs. gefitinib)					
Erlotinib	0.91 [0.56–1.46]	0.83 [0.48–1.43]	0.88 [0.54–1.42]	0.87 [0.48–1.56]	0.85 [0.52–1.39]
Afatinib	0.63 [0.37–1.08]	0.61 [0.34–1.10]	0.63 [0.37–1.08]	0.65 [0.36–1.19]	0.63 [0.37–1.08]
Osimertinib	†	†	†	†	†
Anti-VEGF treatment (vs. no anti-VEGF)	0.28 [0.04–1.99]	0.35 [0.05–2.53]		0.30 [0.04–2.24]	
Male (vs. female)	1.52 [1.03–2.26] *	1.57 [1.04–2.36] *	1.63 [1.09–2.44] *	1.53 [1.00–2.34] *	1.54 [1.03–2.29] *
Age (≥65 vs. <65)	1.32 [0.91–1.91]	1.43 [0.95–2.14]	1.45 [0.99–2.12]	1.48 [0.98–2.24]	1.42 [0.97–2.08]
ECOG PS (≥2 vs. ≤1)	0.72 [0.38–1.34]	0.70 [0.36–1.37]	0.65 [0.34–1.22]	0.81 [0.41–1.59]	
Exon 19 deletion vs. L858R	1.10 [0.76–1.59]	1.10 [0.74–1.64]		1.18 [0.79–1.77]	
PD–L1 ≥ 1% vs. PD-L1 < 1% or not tested	1.17 [0.79–1.72]	1.20 [0.78–1.84]		1.20 [0.77–1.88]	
Number of brain metastases (≥5 vs. <5)	0.88 [0.60–1.30]	0.97 [0.63–1.47]		0.91 [0.59–1.39]	
Extracranial metastasis (vs. no)	1.69 [0.97–2.93]	1.92 [1.08–3.43] *	1.99 [1.14–3.49] *		
Lung metastasis (vs. no)	1.36 [0.94–1.98]			1.23 [0.81–1.88]	
Pleural metastasis/effusion (vs. no)	1.13 [0.78–1.64]			1.23 [0.77–1.97]	
Bone metastasis (vs. no)	1.38 [0.94–2.03]			1.52 [0.98–2.37]	1.43 [0.97–2.11]
Liver metastasis (vs. no)	0.82 [0.45–1.49]			0.66 [0.35–1.28]	
Pericardial metastasis/effusion (vs. no)	0.86 [0.56–1.32]			0.64 [0.37–1.10]	
Adrenal metastasis (vs. no)	1.24 [0.70–2.17]			1.19 [0.61–2.31]	
Other site metastasis (vs. no)	1.24 [0.57–2.67]			1.07 [0.43–2.66]	

Abbreviations: ECOG, Eastern Cooperative Oncology Group; EGFR, epidermal growth factor receptor; OP, surgery for brain tumor; PD-L1, programmed death ligand 1; PS, performance status; RT, radiotherapy for brain tumor; TKI, tyrosine kinase inhibitor; VEGF, vascular endothelial growth factor. A patient with cancer harboring both the exon 19 deletion and L858R was included in the exon 19 deletion group. Data are presented as hazard ratio (HR) with 95% confidence interval. After building the maximal models of multivariable Cox regression (models 3 and 4), the corresponding reduced multivariable Cox regression models (models 3R and 4R, respectively) were built using backward variable selection, retaining only variables with *p*-values < 0.15. † HR could not be assessed due to no event in a group. * *p*-value < 0.05.

## Data Availability

Not applicable.

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
