# Peer review of "Different Tyrosine Kinase Inhibitors Used in Treating EGFR-Mutant Pulmonary Adenocarcinoma with Brain Metastasis and Intracranial Intervention Have No Impact on Clinical Outcomes"

_cancers, 2022, doi:10.3390/cancers15010187_

Round 1
Reviewer 1 Report
The manuscript titled 'Different Tyrosine Kinase Inhibitors in Treating EGFR-Mutant Pulmonary Adenocarcinoma with Brain Metastasis and Intracranial Intervention Has No Impact on Clinical Outcome' adopts a retrospective statistical analysis based approach to evaluate the optimal approach to better treat NSCLC patients with brain metastasis. The proposed approach and obtained results are important to the field given the severity of the disease and the low survival rates. While the manuscript is well written and the presented findings are easy to follow and evaluate, there are a few concerns associated with the current manuscript. They are listed below:
1) The authors need to revisit the introduction and better highlight the importance of performing such a retrospective study. Additionally, some of the previously published studies which adopt a similar approach should be highlighted in the introduction. The introduction should better introduce EGFR mutations and should expand on associated TKI treatment strategies which have been previously applied based on specific mutation phenotypes.
2) Since the entire manuscript is built around analyzing previously obtained patient demographic, clinical, and pathological data, the authors need to provide detailed justification for the adopted statistical approaches and provide clear explanations for why the adopted statistical approach is best suited for the study.
3) The discussion section for the manuscript has major issues: While the author's state previously published results from other papers that contradict or differ from current results, the authors never clearly discuss the possible reasons for the observed discrepancies. The discussion should actually explore these observed differences and provide valid explanations to justify the current findings (for example Lines 289-297).
4) The authors state some of the caveats of the study. One major caveat is the extremely small sample size for patients receiving osimertinib treatment. Given this situation, making any comparative conclusions including results for osimertinib treatment across all tables and graphs is incorrect. The authors should discuss how the caveats listed by them could potentially affect the observations.
5) Recent advances in the field of NSCLC treatment have suggested the employment of combinatorial therapies targeting other receptors associated with enhanced metastasis and cancer progression. This aspect is not explored within the manuscript. Receptors like integrins are key molecules contributing to cancer progression and metastasis and attempts are being made in order to develop integrin-targeting anti-cancer drugs. Angiogenesis is another essential factor for tumor progression and metastasis. Integrins play a key role in the regulation of the process of tumor angiogenesis which consists of basement membrane degradation, endothelial cell migration, proliferation, and stabilization. The authors need to explore these new directions and discuss the current results in light of future potential. See suggested references:
a) Slack, R.J., Macdonald, S.J.F., Roper, J.A. et al. Emerging therapeutic opportunities for integrin inhibitors. Nat Rev Drug Discov 21, 60–78 (2022). https://doi.org/10.1038/s41573-021-00284-4
b) , , , et al. Construction of a prognostic risk assessment model for lung adenocarcinoma based on Integrin β family-related genes. J Clin Lab Anal. 2022; 36:e24419. doi:10.1002/jcla.24419
c) Tejeshwar C. Rao, Victor Pui-Yan Ma, Aaron Blanchard, Tara M. Urner, Shreya Grandhi, Khalid Salaita, Alexa L. Mattheyses; EGFR activation attenuates the mechanical threshold for integrin tension and focal adhesion formation. J Cell Sci 1 July 2020; 133 (13): jcs238840. doi: https://doi.org/10.1242/jcs.238840
d) Bergonzini C, Kroese K, Zweemer AJM, Danen EHJ. Targeting Integrins for Cancer Therapy - Disappointments and Opportunities. Front Cell Dev Biol. 2022 Mar 9;10:863850. doi: 10.3389/fcell.2022.863850. PMID: 35356286; PMCID: PMC8959606.
e) Hassanein SS, Abdel-Mawgood AL, Ibrahim SA. EGFR-Dependent Extracellular Matrix Protein Interactions Might Light a Candle in Cell Behavior of Non-Small Cell Lung Cancer. Front Oncol. 2021 Dec 15;11:766659. doi: 10.3389/fonc.2021.766659. PMID: 34976811; PMCID: PMC8714827.
Author Response
To reviewer 1:
Dear reviewer:
Thank you very much for your comments on the manuscript. Based on your comments and request, we have made an extensive modifications on the original manuscript. Your questions were answered below accordingly.
The manuscript titled 'Different Tyrosine Kinase Inhibitors in Treating EGFR-Mutant Pulmonary Adenocarcinoma with Brain Metastasis and Intracranial Intervention Has No Impact on Clinical Outcome' adopts a retrospective statistical analysis based approach to evaluate the optimal approach to better treat NSCLC patients with brain metastasis. The proposed approach and obtained results are important to the field given the severity of the disease and the low survival rates. While the manuscript is well written and the presented findings are easy to follow and evaluate, there are a few concerns associated with the current manuscript. They are listed below:
1. The authors need to revisit the introduction and better highlight the importance of performing such a retrospective study. Additionally, some of the previously published studies which adopt a similar approach should be highlighted in the introduction. The introduction should better introduce EGFR mutations and should expand on associated TKI treatment strategies which have been previously applied based on specific mutation phenotypes.
Reply. We revised the section of introduction according to reviewers’ kind suggestion, thanks
2. Since the entire manuscript is built around analyzing previously obtained patient demographic, clinical, and pathological data, the authors need to provide detailed justification for the adopted statistical approaches and provide clear explanations for why the adopted statistical approach is best suited for the study.
Reply. In this retrospective study, we adopted statistical approaches which were commonly used in similar studies. Categorical variables and continuous variables were compared using the Chi-square test and analysis of variance, respectively. Survival times were estimated with the Kaplan-Meier method, and differences between groups were compared with the log-rank test. Both univariate and multivariable Cox regression analyses were performed to determine the predictive factors for PFS, iPFS, and OS, and hazard ratios (HR) with 95% confidence intervals (CIs) were presented for the predicting factors. To avoid over-adjustment, we used a backward variable selection method, keeping only variables with p values <0.15, to develop reduced multivariable models.
3. The discussion section for the manuscript has major issues: While the author's state previously published results from other papers that contradict or differ from current results, the authors never clearly discuss the possible reasons for the observed discrepancies. The discussion should actually explore these observed differences and provide valid explanations to justify the current findings (for example Lines 289-297).
Reply. Till now, there were no randomized controlled trials to compare lung cancer with brain metastasis. Almost all published studies about lung cancer with brain metastasis trials were retrospective and had small sample sizes. Smaller numbers, different enrolled criteria, different opinions by physicians, and retrospective entities made these studies have discrepancies. We added more discussion in the manuscript and hope our study adds more information to the important and crucial issue.
4. The authors state some of the caveats of the study. One major caveat is the extremely small sample size for patients receiving osimertinib treatment. Given this situation, making any comparative conclusions including results for osimertinib treatment across all tables and graphs is incorrect. The authors should discuss how the caveats listed by them could potentially affect the observations.
Reply. The number of patients receiving Osimertinib in our cohort was relatively small in this study. Taiwan’s National Health Insurance (NHI) is a government-run single-payer program introduced in 1995 that now covers more than 99% of 23 million Taiwanese citizens. only gefitinib, erlotinib, and afatinib were allowed by Taiwan NHI before 2020. Osimertinib was relatively expensive and only some patients could afford osimertinib. Osimertinib was allowed for lung adenocarcinoma with brain metastasis in Deletion 19 subgroup by Taiwan’s NHI since April 2022. This is the reason why only small numbers of lung cancer with brain metastasis were treated with Osimertinib in our study. We will enroll more lung cancer with brain metastasis in future studies.
5. Recent advances in the field of NSCLC treatment have suggested the employment of combinatorial therapies targeting other receptors associated with enhanced metastasis and cancer progression. This aspect is not explored within the manuscript. Receptors like integrins are key molecules contributing to cancer progression and metastasis and attempts are being made in order to develop integrin-targeting anti-cancer drugs. Angiogenesis is another essential factor for tumor progression and metastasis. Integrins play a key role in the regulation of the process of tumor angiogenesis which consists of basement membrane degradation, endothelial cell migration, proliferation, and stabilization. The authors need to explore these new directions and discuss the current results in light of future potential. See suggested references:
a) Slack, R.J., Macdonald, S.J.F., Roper, J.A. et al.Emerging therapeutic opportunities for integrin inhibitors. Nat Rev Drug Discov21, 60–78 (2022). https://doi.org/10.1038/s41573-021-00284-4
b) Wu, Y, Fu, L, Wang, B, et al. Construction of a prognostic risk assessment model for lung adenocarcinoma based on Integrin β family-related genes. J Clin Lab Anal. 2022; 36:e24419. doi:1002/jcla.24419
c) Tejeshwar C. Rao, Victor Pui-Yan Ma, Aaron Blanchard, Tara M. Urner, Shreya Grandhi, Khalid Salaita, Alexa L. Mattheyses; EGFR activation attenuates the mechanical threshold for integrin tension and focal adhesion formation. J Cell Sci1 July 2020; 133 (13): jcs238840. doi: https://doi.org/10.1242/jcs.238840
d) Bergonzini C, Kroese K, Zweemer AJM, Danen EHJ. Targeting Integrins for Cancer Therapy - Disappointments and Opportunities. Front Cell Dev Biol. 2022 Mar 9;10:863850. doi: 10.3389/fcell.2022.863850. PMID: 35356286; PMCID: PMC8959606.
e) Hassanein SS, Abdel-Mawgood AL, Ibrahim SA. EGFR-Dependent Extracellular Matrix Protein Interactions Might Light a Candle in Cell Behavior of Non-Small Cell Lung Cancer. Front Oncol. 2021 Dec 15;11:766659. doi: 10.3389/fonc.2021.766659. PMID: 34976811; PMCID: PMC8714827.
Reply. Receptors like integrins are key molecules contributing to cancer progression, and we added and revised the discussion and cited these important references for the employment of combinatorial therapies targeting other receptors associated with enhanced metastasis and cancer progression. Thanks.
Reviewer 2 Report
1. After line 52, what is your interpretation of your findings? Should intial surgery and/or local therapy be avoided in patients with brain metastases and EGFR mutation positive tumors be avoided? Some considerations include the following the following ideas. Are more real world data needed in this group of patients? Is a randomized study comparing EGFR TKI only versus EGFR TKI plus surgery and/or radiation? Are more data needed with third generation EGFR TKIs?
2. In lines 90 - 92, it seems that your primary objective was to compare outcomes for patients who had resection of brain metastases or brain radiation versus patients who had only EGFR TKI.
3. In lines 126 - 130, please include some comments about frequency of and type of brain imaging. Also, please include some description of frequency of CT scans or PET scans for monitoring extra-cranial disease sites.
4. In lines 285 - 297, suggest including reference to and description of findings of publication by Reungwetwattana T(J Clin Oncol 36: 1320 - 1327,2018). This paper described comparison of osimertinib versus either gefitinib or erlotinib for treatment of brain metastases.
5. Also would emphasize that the number of patients treated with osimertinib in your study was small.
6. In lines 326 - 334, suggest including comments regarding the implications of your real world findings for treatment of patients with brain metastases and in tumors containing EGFR mutations.
Author Response
To reviewer 2
Dear reviewer:
I am very grateful for your comments on the manuscript. According to your advice, we amended the relevant part of the manuscript. Your question was answered below.
1., After line 52, what is your interpretation of your findings? Should initial surgery and/or local therapy be avoided in patients with brain metastases and EGFR mutation positive tumors be avoided? Some considerations include the following the following ideas. Are more real world data needed in this group of patients? Is a randomized study comparing EGFR TKI only versus EGFR TKI plus surgery and/or radiation? Are more data needed with third generation EGFR TKIs?
Reply. In our study, Intracranial intervention had no statistically significant impact on the treatment outcomes of patients with EGFR mutations and brain metastasis who received EGFR TKIs as first-line therapy. In daily practice, many lung cancers with brain metastasis received intracranial intervention such as craniotomy and radiotherapy. However, a craniotomy may have some severe fatal complications such as intracranial hemorrhage. In addition, radiotherapy to the brain always has poor cognition months later. Although the treatment decision depends on the neurogenic symptom and the decision-making of the physicians and patients, our study suggested that intra-cranial intervention was not necessary for this kind of patient. Further randomized control trials are urgently needed to validate the findings.
2. In lines 90 - 92, it seems that your primary objective was to compare outcomes for patients who had resection of brain metastases or brain radiation versus patients who had only EGFR TKI.
Reply. Yes, as previously mentioned, in our cohort, we try to reveal these patients underwent unnecessary intracranial intervention, these processes may have some severe complications. Our study preferred EGFR TKI alone to a combination of intracranial intervention to EGFR TKI, and we compared the efficacy of different EGFR TKIs in mutant-EGFR lung cancer with brain metastasis. In addition, we also compare the different efficacy in different EGFR TKIs.
3. In lines 126 - 130, please include some comments about frequency of and type of brain imaging. Also, please include some description of frequency of CT scans or PET scans for monitoring extra-cranial disease sites.
Reply. The majority of these enrolled patients with brain metastasis often received 3-6 months intervals to check brain tumors by brain MRI in our daily practices. However, because of the retrospective entity, the frequency, and type of brain imaging and CT scans depend on the decision of physicians and the condition and symptoms of the patients, therefore, the check-up interval of the brain image not being clearly defined.
4. In lines 285 - 297, suggest including reference to and description of findings of publication by Reungwetwattana T(J Clin Oncol 36: 1320 - 1327,2018). This paper described a comparison of osimertinib versus either gefitinib or erlotinib for treatment of brain metastases.
Reply. We cite the very important reference of publication by Reungwetwattana T(J Clin Oncol 36: 1320 - 1327,2018) to make the manuscript more reliable and readable.
5. Also would emphasize that the number of patients treated with osimertinib in your study was small.
Reply. The number of patients receiving Osimertinib in our cohort was relatively small size in this study. Taiwan’s National Health Insurance (NHI) is a government-run single-payer program introduced in 1995 that now covers more than 99% of 23 million Taiwanese citizens. only gefitinib, erlotinib, and afatinib were offered by Taiwan NHI before 2020. Osimertinib was very expensive and was allowed for lung adenocarcinoma with brain metastasis in Deletion 19 subgroup by Taiwan’s NHI since April 2022. This is the reason why only a small number of lung cancer with brain metastasis are treated with osimertinib. We will enroll more lung cancer with brain metastasis in future studies.
5. In lines 326 - 334, suggest including comments regarding the implications of your real-world findings for the treatment of patients with brain metastases and in tumors containing EGFR mutations.
Reply. Thanks, we revised it in the manuscript and wish our findings let physicians who treat these poor lung cancers a good reference.
Round 2
Reviewer 1 Report
The authors have addressed all concerns raised in the first round of review. The manuscript is significantly improved and presents the research of importance to the field. The improvements to the manuscript have made it ready for publication.